# The Effect of *Mucuna pruriens* on Depression-like Behavior Induced by a Mild Traumatic Brain Injury in Rats Is Associated with a Decrease in Brain Nitrite and Nitrate Levels

**DOI:** 10.3390/neurosci6040092

**Published:** 2025-09-24

**Authors:** Alfonso Mata-Bermudez, Ricardo Trejo-Chávez, Marina Martínez-Vargas, Adán Pérez-Arredondo, Araceli Diaz-Ruiz, Camilo Rios, Héctor Alonso Romero-Sánchez, María de los Ángeles Martínez-Cárdenas, Perla Ugalde-Muñiz, Roxana Noriega-Navarro, Luz Navarro

**Affiliations:** 1Departamento de Fisiología, Facultad de Medicina, Universidad Nacional Autónoma de México, Mexico City 04510, Mexico; alfonsomata24@hotmail.com (A.M.-B.); rtrejochavez@gmail.com (R.T.-C.); marina.martinez@facmed.unam.mx (M.M.-V.); aperez@facmed.unam.mx (A.P.-A.); perlifa@hotmail.com (P.U.-M.); roxnn77@gmail.com (R.N.-N.); 2Departamento de Atención a la Salud, Universidad Autónoma Metropolitana Unidad Xochimilco, Mexico City 04510, Mexico; alonsoromero1278@gmail.com (H.A.R.-S.); amartinez@correo.xoc.uam.mx (M.d.l.Á.M.-C.); 3Programa de Doctorado en Ciencias Biomédicas, Universidad Nacional Autónoma de México, Mexico City 04510, Mexico; 4Programa de Maestría y Doctorado en Ciencias Médicas, Odontológicas y de la Salud, Investigación Clínica Experimental, Universidad Nacional Autónoma de Méxic, Mexico City 04510, Mexico; 5Departamento de Neuroquímica, Instituto Nacional de Neurología y Neurocirugía Manuel Velasco Suarez, Mexico City 14269, Mexico; adiaz@innn.edu.mx; 6Instituto Nacional de Rehabilitación, Luis Guillermo Ibarra Ibarra, Mexico City 14389, Mexico; camrios@yahoo.com.mx; 7Departamento de Sistemas Biológicos, Universidad Autónoma Metropolitana Unidad Xochimilco, Mexico City 04510, Mexico

**Keywords:** head injury, depression, *Mucuna pruriens*, oxidative stress, nitric oxide

## Abstract

Traumatic brain injury (TBI), even when mild, has been associated with the presence of depression. Depression is a mood disorder characterized by persistent negative thoughts and sadness and is challenging to treat due to the multiple mechanisms involved in its pathophysiology, including increased nitric oxide (NO) levels. There are no completely safe and effective pharmacological strategies to treat this disorder. *Mucuna pruriens* (MP) has been shown to possess neuroprotective properties by regulating inflammatory responses and nitric oxide synthase activity. In this study, we evaluated the antidepressant-like effect of MP in male Wistar rats with induced mild traumatic brain injury (mTBI). MP extract (50 mg/kg i.p.) was administered immediately after mTBI and every 24 h for five days. We used the rats’ preference for sucrose consumption to assess the presence of depression-like behavior and analyzed the nitrite and nitrate levels in their cerebral cortex, striatum, midbrain, and nucleus accumbens. Untreated animals with mTBI showed a reduced preference for sucrose than those treated with MP, whose preference for sucrose was similar to that of sham animals. Increased nitrite and nitrate levels were observed in different brain regions in the TBI subjects; however, this increase was not observed in MP-treated animals. MP reduces behavior associated with depression and the brain NO levels in rats with mTBI.

## 1. Introduction

TBI is considered a public health issue, as it is one of the leading causes of death and disability worldwide [1]. TBI is defined as the total or partial disruption of normal brain cell functions or other brain pathologies arising due to an external mechanical force, leading to impaired physical, psychological, and cognitive functions [2,3,4]. Depending on the severity of the lesion (mild, moderate, or severe), various types of secondary biochemical damage occur, including inflammation, apoptosis, oxidative stress, and other pathophysiological complications [5]. The most prevalent TBI type is mTBI, which accounts for around 55 million cases yearly [6].

The psychiatric symptoms commonly associated with TBI can vary depending on the type of injury, location, and amount of brain tissue affected [7] and include cognitive problems (deficits in memory and executive functions) and problems with affective behavior (depression and anxiety) [8]. The prevalence of depressive disorders arising from TBI and even mTBI has been reported to range from 10 to 77%, representing a considerable reduction in social functioning and quality of life for patients [9]. Depression is a chronic psychiatric condition characterized by a depressed mood, social withdrawal, and anhedonia [10]. The current therapeutic strategies use selective serotonin reuptake inhibitors (sertraline or citalopram) as the first line of treatment [11]. However, there is still little evidence that pharmacological treatments are effective for this condition after TBI [12].

On the other hand, it has been reported that NO, a retrograde gaseous neurotransmitter involved in the activation of various cell signaling pathways, plays a fundamental role in the pathogenesis of depression. A recent study reported that depressive behaviors are associated with an increase in NO levels in obese mice [13]; in addition, glutamatergic modulators, including N-methyl-D-aspartate (NMDA) receptor antagonists, are known to exhibit similar effects on NO levels to those observed with antidepressants in behavioral models of mild chronic stress [14]. The entry of calcium through NMDA receptors activates the enzyme NO synthase (NOS), converting L-arginine into NO and L-citrulline [15,16]. There are various isoforms of this enzyme, such as inducible NOS (iNOS), endothelial NOS (eNOS), and neuronal NOS (nNOS) [17]. It has been reported that after TBI, NO synthesis increases in the brain, as well as the levels of superoxide anion, favoring the formation of peroxynitrites (ONOO^−^), some of the most reactive molecules. They alter the structure of the brain and function of various biomolecules, which translates into irreversible damage leading to cell death, vascular dysfunction, and posttranslational protein modifications [18].

In the search for new potential therapeutic alternatives, interest has increased in studying medicinal plants, such as the legume MP, which belongs to the Fabaceae family [19]. Previous reports have indicated that hydroalcoholic MP extract (100 µg/mL) reduces the NO concentration in RAW 264.7 macrophages stimulated by lipopolysaccharide [20]. Likewise, the administration of a hydroalcoholic MP extract (100 mg/kg, p.o.) for seven days decreased the NO concentration and iNOS expression in the striatum of mice with Parkinson’s disease induced by paraquat [21]. Furthermore, we recently published a review on the possible use of MP as a treatment for depression [22]. In addition, in a previous report, we showed that short-term treatment with MP prevented depression-like behavior following a mild TBI [23]. In the present work, we expand on this and show that MP extract prevents an increase in nitrites and nitrates caused by mTBI in various brain areas.

## 2. Materials and Methods

### 2.1. Animals

Adult male Wistar rats with a body weight of 250–300 g were housed under controlled vivarium conditions with a 12/12 h light/dark cycle, with the lights turned on at 7:00 h, and had free access to water and food. Surgical procedures and behavioral tests were performed in accordance with both the Mexican Official Standard NOM-062-ZOO-1999 [24] and the international Guidelines for the Care and Use of Laboratory Animals provided by the NIH, USA. The animals were obtained from the Bioterium Unit of the Faculty of Medicine at the National Autonomous University of Mexico. All the experimental procedures were approved by the Ethics Committee of the Faculty of Medicine of the National Autonomous University of Mexico (UNAM) (project: 031-CIC-2019).

### 2.2. Drugs

Lyophilized MP extract was provided by Biofab México (Irapuato, Guanajuato, Mexico). The extract contained 56% L-dopa. Its other main components were arginine, stizolamine, and the fructo-oligosaccharides sucrose and nystose [25]. A 50 mg/kg dose of the lyophilized extract was prepared in a 0.9% saline solution [26].

### 2.3. Experimental Design

Male Wistar rats were randomly distributed into twenty experimental groups, including a naïve group (N); a sham group (which underwent a simulated surgical procedure, S); a group with mTBI treated with a vehicle (0.9% saline solution, SS) every 24 h for five days (V); and a group with lesions treated with the lyophilized MP extract at a dose of 50 mg/kg i.p. daily for five days, starting 15 min after the injury (MP). The occurrence of depression-like behavior was assessed after the mTBI at 3, 7, 15, 30, and 60 days. For each group, once the behavioral test had been completed and anesthesia had been administered (50 mg/kg of sodium pentobarbital i.p.), the animals were killed through decapitation, and their striatum, cerebral cortex, nucleus accumbens, and midbrain were extracted to determine the concentrations of nitrites and nitrates present.

### 2.4. Surgical Procedure

Under aseptic conditions and anesthesia (xylazine (0.26 mg/kg, i.p.), ketamine (66 mg/kg, i.p.), and acepromazine (1.3 mg/kg, i.p.)), all the animals in the S, V, and MP groups were placed individually into a stereotaxic device (Stoelting Co., Wood Dale, IL, USA) to induce mTBI as previously reported [27]. The closed head injury (CHI) model was used, and a 1 mm diameter calibrated pneumatic piston was used to impact each rat’s skull with a force of 20 lb and depth of 4 mm. The impact occurred at −2 mm anterior–posterior (AP) and +1.4 mm lateral (L) stereotaxic coordinates, which corresponded to the primary motor cortex, M1 [28].

### 2.5. Sucrose Preference Test

To evaluate the occurrence of depressive-like behavior induced by mTBI, the preference for sucrose consumption was determined in groups of 12 subjects. Each experimental subject was placed individually into a box. Two drinking bottles, one containing 150 mL of water at room temperature and the other containing 150 mL of a 2% sucrose solution, were positioned at each end of the box for two hours. The positions of the drinking bottles were switched after one hour to avoid a position bias, and at the end of the test, the drinking bottles were removed. The ingested water and sucrose solution were quantified as reported by Liu et al. [29]. A reduction in the consumption of sugar water would demonstrate depressive-like behavior.

### 2.6. Quantification of Nitrates and Nitrites Using the Griess Method

The concentrations of nitrites and nitrates were quantified using the Griess reaction, which quantifies the stable metabolites of nitric oxide. Brain tissues (striatum, cerebral cortex, nucleus accumbens, and midbrain) were weighed and homogenized in phosphate-buffered saline (PBS, pH 7.4) and deproteinized with zinc sulfate (ZnSO_4_). After centrifugation, 100 µL of the supernatant was transferred to 96-well plates and mixed with 200 µL of freshly prepared Griess reagent, consisting of vanadium (III) chloride, N-(1-naphthyl) ethylenediamine dihydrochloride, and sulfanilamide. Samples were incubated for 30 min at 37 °C, and the absorbance was measured at 540 nm using a microplate reader (BIOTEK ELX808, BioTek Instruments, Inc., Winooski, VT, USA). Quantification by triplicate was performed against a sodium nitrate calibration curve prepared under identical conditions [30].

### 2.7. Statistical Analysis

In all cases, an exploratory analysis of the data was performed to determine whether the variables showed a normal distribution and homogeneity of variances. For this, the Kolmogorov–Smirnov and Levene statistical tests were used. Next, we conducted parametric statistical tests. Significant differences among the groups were determined using a two-way analysis of variance, followed by the two-stage linear step-up procedure proposed by Benjamin, Krieger, and Yekutieli, in the statistical program GraphPad Prism version 9.3.1 for Windows. This procedure was employed to control the false discovery rate (FDR) across multiple comparisons, including those between the different brain regions and time points analyzed. We selected this method because it provides a balanced approach to limiting false positives while preserving statistical power. We have included the predicted (LS) mean difference, q value, and individual P value for each pair of compared means in the Appendix A.

## 3. Results

### 3.1. Treatment with a Lyophilized Mucuna pruriens Extract Decreases Depressive-like Behavior After Mild TBI in Rats

The results obtained regarding the preference for sucrose consumption after short-term treatment with a lyophilized MP extract (15 min after TBI, then once a day for five days) in rats with mTBI are shown in Figure 1. The results obtained indicate that the animals with mTBI that only received the vehicle (0.9% SS) consumed a significantly (*p* < 0.05) lower percentage of sugar water than the rats that underwent a simulated surgical procedure (sham) from day seven to sixty post-mTBI and the rats that did not undergo any manipulation (naïve) from day 15 to 60. Likewise, short-term treatment with a lyophilized extract from MP plants (50 mg/kg, i.p.) significantly prevented the decrease in sucrose consumption from day 7 to 60 post-mTBI (*p* < 0.05) seen in the group of injured animals receiving the vehicle (0.9% SS). The findings suggest that short-term treatment with a lyophilized MP extract prevents the development of depressive-like behavior, resulting in amounts of sucrose consumption close to those of the naïve and sham control groups.

### 3.2. Treatment with a Lyophilized Mucuna pruriens Extract Decreases the Nitrite and Nitrate Levels in a Rat Model of Mild TBI

Figure 2 shows the concentrations of nitrites and nitrates found in different brain regions. Panel A shows the cerebral cortex, Panel B the striatal nucleus, Panel C the midbrain, and Panel D the nucleus accumbens from rats with mTBI treated with a vehicle or repeated doses of MP starting 15 min after TBI, with one dose given every 24 h for five days.

The results showed that mTBI increased the nitrite and nitrate concentrations in the motor cortex (Figure 2A) from day 7 post-TBI to day 30 compared with those in naïve or sham-treated rats. Treatment with MP significantly prevented this increase (*p* < 0.05).

In the striatum (Figure 2B), a significant increase in nitrites and nitrates (*p* < 0.05) was observed in the group of injured animals compared to the naïve and sham groups on days 3, 7, and 30 after mTBI; on days 15 and 60, these values did not reach statistical significance. However, the group of animals treated with MP (Group M) did not exhibit (*p* < 0.05) these increases.

In the midbrain (Figure 2C), we observed that mTBI induced a significant increase in the concentration of nitrites and nitrates on days 3, 7, 15, and 30 post-mTBI when compared with those in Groups N and S. We could also see that administering the lyophilized MP extract (M) prevented this significant increase (*p* < 0.05) from day 7 to 30 after mTBI, in contrast to the results for Group V.

Regarding the nucleus accumbens (Figure 2D), an increase (*p* < 0.05) in the concentration of nitrites and nitrates after mTBI was observed from day 3 to 60 post-mTBI; this increase was prevented by the administration of the lyophilized MP extract (M) from day 7 to 60 post-mTBI.

## 4. Discussion

Behaviors associated with depression frequently occur after mTBI [9]. Since there is still no treatment that is entirely effective for this condition, exploring new therapeutic alternatives using various natural products has gained popularity. Recently, it has been reported that MP can decrease the occurrence of behaviors associated with depression by reducing oxidative stress in various animal models [21,31,32,33], suggesting that it could be a promising treatment for managing depressive behaviors in patients with TBI. The results obtained in this mTBI model indicate that treatment with a lyophilized MP extract can prevent a decrease in the sucrose consumption preference induced by mTBI from day 7 to day 60 post-mTBI (Figure 1). It is worth noting that the effect on depression-like behavior observed in the MP group was very similar to our previous results on the immobility time in a forced swim test [23]. Behavioral models where consumption of or a preference for sweet solutions is evaluated are associated with the initiation, maintenance, and regulation of a loss of interest in activities that were previously considered pleasurable (anhedonia), which is associated with depression [34,35]. The antidepressant effects induced by MP may be related to its antioxidant effect and ability to strengthen antioxidant defenses [31,36]. Various reports indicate that MP seed extract (12.5, 25, and 50 μg/mL) reduces the concentration of reactive oxygen species (ROS) and reactive nitrogen species (RNS) in in vitro [37] and in vivo models [38].

Additionally, it has been reported that MP treatment suppresses the expression of proinflammatory cytokines and enzymes, including iNOS, in mice with Parkinson’s disease [39]. Although mechanisms causing depressive behaviors after TBI are not fully understood, evidence suggests that depression occurs due to abnormal functioning in the metabolism of various neurotransmitters, such as dopamine and serotonin; however, it has been observed that oxidative stress and neuroinflammation also play a crucial role [40]. The formation of RNS, such as NO and its derivative compounds, including ONOO^−^, is increased in depressed patients [41,42] and those with TBI [43]. NO is a free radical that plays a fundamental role in the pathophysiology of major depression [44]. In support of the above, a considerable increase in NO concentrations has been reported in depressed and suicidal patients [45]. A TBI-induced increase in NO concentrations may facilitate inflammatory processes in the brain by inducing a massive increase in proinflammatory cytokines, such as interleukin 1-beta (IL-1β) and tumor necrosis factor-alpha (TNF-α), over prolonged periods [46,47]. This inflammatory response activates signaling cascades that can activate constitutive NOS enzymes (eNOS and nNOS) and induce upregulation of iNOS (independent of calcium) [18], facilitating glutamatergic signaling and inducing neuronal death [46]. It is also known that NO is related to the modulation of cyclic guanosine monophosphate (cGMP) levels, producing a depression-like state in animals [48]. Likewise, the data obtained in our study indicate that the treatment used reduced the concentration of nitrites and nitrates (Figure 2) in the cerebral cortex, striatum, midbrain, and nucleus accumbens from 7 days after mTBI compared to that in the group of injured rats that only received the vehicle. The results obtained suggest that in rats with mTBI, treatment with a lyophilized MP extract prevents the appearance of behaviors associated with depression by reducing the concentrations of nitrites and nitrates in various brain regions related to the development of depressive behaviors (Figure 1). The findings are consistent with previous studies, which indicate that treatment with coumarin compounds, such as auraptene (7-geranyloxycoumarin), can decrease NO production by reducing the iNOS activity. This can prevent the onset of behaviors associated with depression in rodents [49]. Likewise, treatment with nonselective inhibitors of NOS (Nω-nitro-l-arginine methyl ester) and those selective for nNOS (7-nitroindazole) and iNOS (N-([3-(aminomethyl)phenyl]methyl) ethanimidamide dihydrochloride) decreased the immobility time in a forced swimming test, an effect similar to that achieved with antidepressant drugs such as venlafaxine or fluoxetine [50,51,52].

In the model of mild TBI used in the present work, the antidepressant effect and decrease in the nitrite and nitrate concentrations induced by short-term treatment with the lyophilized MP extract could have been related to interaction with the dopaminergic system [53]. The mesolimbic and mesocortical dopaminergic systems are involved in anhedonia and a lack of motivation, two core symptoms of depression [54]. The inflammatory processes induced by TBI increase the concentration of NO and contribute to the oxidation of tetrahydrobiopterin (BH_4_), a necessary cofactor for the conversion of phenylalanine to tyrosine and tyrosine to L-3,4-dihydroxyphenylalanine (L-DOPA). These are necessary for the synthesis of dopamine, which in turn is a cofactor for the formation of NOS [55]. The information obtained suggests that MP prevents the formation of nitrites and nitrates in the brain nuclei that comprise the mesocortical and mesolimbic pathways. In support of the above, reports indicate that oral treatment with MP is capable of increasing the dopamine concentrations in the cerebral cortex [56], preventing a decrease in tyrosine hydroxylase (TH)’s immunoreactivity in the striatum, and increasing the iNOS expression observed in an animal model of Parkinson’s disease [57].

Based on our results, it is worth highlighting that short-term administration of MP extract promotes sustained behavioral recovery after mTBI. Previous evidence indicating that brief pharmacological interventions can produce long-lasting effects is scarce. Nevertheless, some preclinical studies have shown that transient treatments may yield durable neuroprotective and functional benefits: for example, a 7-day cannabidiol (CBD) pretreatment reduced the glutamate excitotoxicity and improved sensorimotor function for up to 30 days post-injury [58]; short-term kahweol dosing attenuated brain tissue loss and enhanced the neurobehavioral outcomes for up to 28 days [59]; and acute TRPM2 channel inhibition improved memory and synaptic plasticity 30 days after TBI [60]. Clinical studies present contradictory findings, while systematic reviews indicate that some agents (e.g., statins, N-acetylcysteine, Cerebrolysin) show possible neuroprotective effects and improved functional outcomes. However, no single short-term pharmacological therapy has been unequivocally proven to improve TBI outcomes in humans [61,62]. Most acute-phase pharmacological trials in adults have shown limited or no effect, with some exceptions for specific agents or patient subgroups [63]. Within this framework, MP emerges as a promising candidate for adjunctive therapy in recovery from TBI.

Finally, although the dopaminergic restoration produced by the 56% L-DOPA content was likely a major driver of the behavioral improvement, we do not attribute the effects solely to L-DOPA. The MP extract used in this study also contained arginine, alkaloids (e.g., stizolamine), and fructo-oligosaccharides (sucrose/nystose), which can act synergistically [25]. Arginine may modulate the arginase–NOS axis and NOS coupling, thereby limiting iNOS-driven nitrosative stress [64,65]. Alkaloids and related phytochemicals from MP have been reported to exert antioxidant and anti-inflammatory effects, contributing to cytoprotection beyond their role as dopamine precursors [21,57]. In addition, fructo-oligosaccharides can act as prebiotics, modulating gut–brain immune signaling and attenuating neuroinflammatory responses [66]. Together, these constituents may provide a multimodal mechanism—enhancing the dopaminergic tone while attenuating neuroinflammation and nitrosative pathways—which is consistent with our findings of reduced nitrite/nitrate levels and decreased depressive-like behavior when MP extract was used as a treatment after TBI.

## 5. Limitations

This study has some limitations that should be acknowledged. First, although it extends our previous work by confirming the long-term antidepressant-like effects of short-term MP administration and also providing new insight into MP’s potential mechanisms of action through modulation of nitrite/nitrate levels, we did not evaluate the toxicity parameters of MP in this study. Nonetheless, previous reports indicate that the dose used (50 mg/kg) was below the levels associated with toxicity in rodents [67], suggesting a favorable safety profile. Second, due to ethical and logistical restrictions, we focused on comparing MP-treated rats with vehicle and sham groups; however, we propose that future studies should include standard antidepressants as comparators. Third, our mechanistic analysis was limited to the quantification of nitrite and nitrate. While the level of these metabolites is widely accepted as an indirect indicator of total NO production, as they represent the stable end products of NO oxidation catalyzed by any NOS isoform [68,69,70,71,72], it should be noted that the nitrite/nitrate levels do not necessarily correlate with the NOS expression [68,73]. Importantly, there is no single marker that fully captures the complexity of NO signaling, and a multimodal approach is recommended. Therefore, future studies should incorporate the assessment of NOS isoforms’ expression and activity, as well as downstream targets such as guanylate cyclase, to examine the relationship between inflammatory cytokines and related changes in neurotransmitters. This approach will provide a more comprehensive understanding of the mechanisms underlying the neuroprotective and antidepressant-like effects of MP.

## 6. Conclusions

Based on the findings of this study, we can conclude that treatment with MP decreases the occurrence of depressive-like behavior observed after TBI, as well as the concentrations of nitrites and nitrates in the cerebral cortex, striatum, midbrain, and nucleus accumbens, demonstrating an association between the levels of NO and depression observed after TBI. This was a safe and effective treatment, as no adverse effects were observed that could put the lives of animals at risk. Based on these results, we suggest using MP as an antidepressant in patients with TBI. However, as noted in the Limitations, further studies are needed, especially on the translational potential and safety of MP for use in humans.

## Figures and Tables

**Figure 1 neurosci-06-00092-f001:**
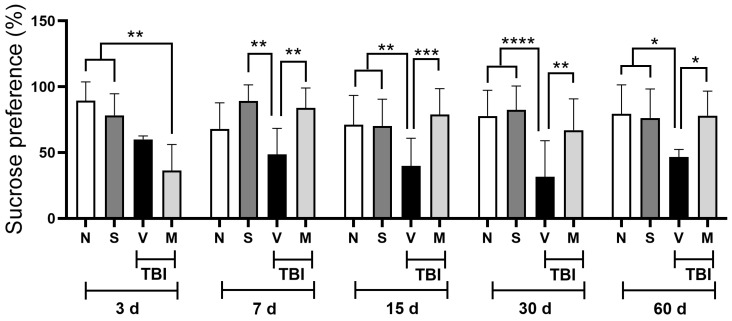
Effect of *Mucuna pruriens* treatment on the sucrose consumption preference in rats posterior to mild traumatic brain injury. The effect of short-term administration (once a day for five days) of *Mucuna pruriens* (50 mg/kg, i.p.) on the percentage of sucrose consumed by the different groups of experimental animals is shown. N: naïve rats; S: sham-operated rats; V: rats with mild TBI and administered a vehicle; M: rats with mild TBI and administered *Mucunna pruriens* extract. All groups were evaluated at 3, 7, 15, 30, and 60 days post-injury. The values are expressed as the mean ± the standard deviation for 12 animals per group. According to a two-way ANOVA, followed by the two-stage linear step-up procedure proposed by Benjamini, Krieger, and Yekutieli, * *p* < 0.05; ** *p* < 0.01; *** *p* < 0.001; and **** *p* < 0.0001. To ensure the clarity of the graph, only significant differences within the same period of time are shown.

**Figure 2 neurosci-06-00092-f002:**
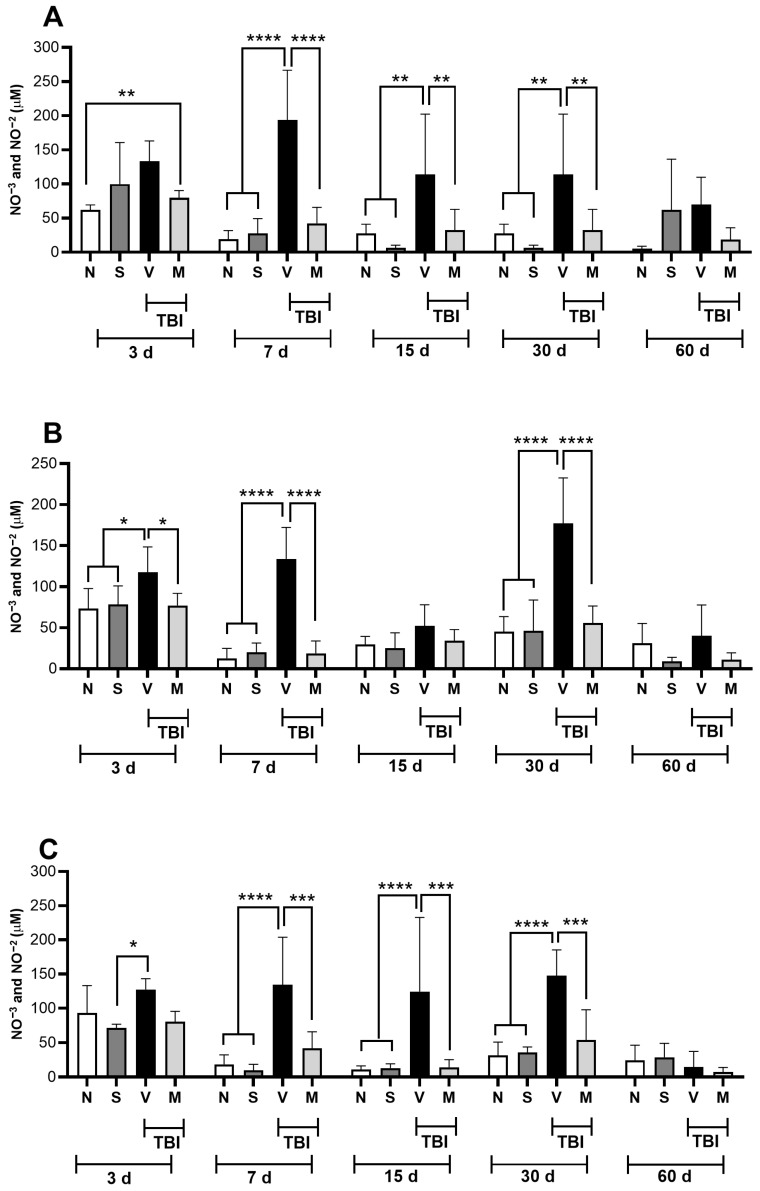
Effect of short-term *Mucuna pruriens* extract administration on the nitrite and nitrate levels in injured tissue in rats posterior to mild traumatic brain injury (mTBI). The nitrite and nitrate levels in the motor cortex (**A**), striatum nucleus (**B**), midbrain (**C**), and nucleus accumbens (**D**) are shown for naïve rats (N), sham-operated rats (S), and rats with mild TBI that received a vehicle (V) or the lyophilized extract of *Mucuna pruriens* (50 mg/kg, i.p., once a day for five days) (M) 3, 7, 15, 30, and 60 days after mild TBI. The values are expressed as the mean ± the standard deviation for 6 animals per group. * *p* < 0.05; ** *p* < 0.01; *** *p* < 0.001; **** *p* < 0.0001 according to a two-way ANOVA, followed by the two-stage linear step-up procedure proposed by Benjamini, Krieger, and Yekutieli. To ensure the clarity of the graph, only significant differences within the same period of time are shown.

## Data Availability

The raw data supporting the conclusions of this article will be made available by the authors, without undue reservation, to any qualified researcher.

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
