# Peer review of "The Effect of Mucuna pruriens on Depression-like Behavior Induced by a Mild Traumatic Brain Injury in Rats Is Associated with a Decrease in Brain Nitrite and Nitrate Levels"

_neurosci, 2025, doi:10.3390/neurosci6040092_

Round 1
Reviewer 1 Report
Comments and Suggestions for Authors
The article investigates the antidepressant-like effect of Mucuna pruriens (MP) in a male Wistar rat model of mild traumatic brain injury (mTBI). The authors report that MP treatment ameliorated depression-like behavior in the sucrose preference test and reduced nitrite and nitrate levels in the cerebral cortex, striatum, midbrain, and nucleus accumbens. Animals with mTBI showed reduced sucrose preference, whereas MP-treated animals displayed levels comparable to sham controls. Similarly, elevated nitrite and nitrate levels observed in various brain regions after mTBI were normalized by MP treatment. With these findings, the study concludes that MP decreases depressive-like behavior and nitric oxide–related metabolites following TBI, suggesting an association between NO levels and post-TBI depression. The authors further note that MP treatment was safe, with no adverse effects observed, and propose MP as a potential antidepressant for TBI patients, while emphasizing the need for further studies. The overall quality of the article should be improved, and the following changes are required.
Major comments:
1. The sucrose preference test was conducted at multiple time points post-TBI (Days 3, 7, 15, 30, and 60) for assessing the progression of depressive-like behavior. However, motor function impairments following TBI are typically more pronounced at early time points and may recover over time. Therefore, motor deficits at earlier stages (e.g., Days 3 and 7) could potentially influence sucrose consumption, whereas at later stages (e.g., Days 30 and 60), the impact of motor impairment on behavior might be minimal.
It would strengthen the study to include motor function assessments to ensure that decreases in sucrose preference are not confounded by motor deficits. If motor data are unavailable, the authors should discuss this limitation and consider its potential influence on the interpretation of behavioral outcomes, particularly during the early post-injury phase.
2. In Figure 2, panels (a) and (b) appear to show identical graphs. Could the authors please verify whether this is correct or if there was an error in figure preparation? Clarification or correction would improve the clarity of the manuscript.
3. While the discussion provides a comprehensive explanation of the potential mechanisms involving nitric oxide (NO) in mediating the antidepressant effects of Mucuna pruriens, the current study presents only nitrite and nitrate level data as indirect indicators. To strengthen the mechanistic insights, it is recommended that at least one direct measure related to NO signaling such as NOS enzyme expression or activity, inflammatory cytokine levels, or related neurotransmitter changes be included as experimental results.
Minor comments:
1. Post-TBI modeling weight loss and the consequent reduction in sucrose intake could potentially confound the interpretation of the sucrose preference test results. It would be important for the authors to report whether body weight was monitored throughout the study to ensure that changes in sucrose consumption were not simply due to weight loss or reduced general intake following TBI. Including this information or discussing it as a limitation would strengthen the reliability of the behavioral findings.
2. On page 6, it would be preferable to use "post-mTBI" instead of "postmTBI" for consistency and clarity.
3. On page 8, while the conclusion appropriately notes that further studies are warranted, it would be helpful and more informative to specify what types of additional research are needed to strengthen the manuscript.
Author Response
We thank the reviewer for the constructive comments that have helped us improve the quality and clarity of the manuscript. Below we provide detailed point-by-point responses. The reviewers’ comments are paraphrased in italics, followed by our responses.
Comments and Suggestions for Authors
The article investigates the antidepressant-like effect of Mucuna pruriens (MP) in a male Wistar rat model of mild traumatic brain injury (mTBI). The authors report that MP treatment ameliorated depression-like behavior in the sucrose preference test and reduced nitrite and nitrate levels in the cerebral cortex, striatum, midbrain, and nucleus accumbens. Animals with mTBI showed reduced sucrose preference, whereas MP-treated animals displayed levels comparable to sham controls. Similarly, elevated nitrite and nitrate levels observed in various brain regions after mTBI were normalized by MP treatment. With these findings, the study concludes that MP decreases depressive-like behavior and nitric oxide–related metabolites following TBI, suggesting an association between NO levels and post-TBI depression. The authors further note that MP treatment was safe, with no adverse effects observed, and propose MP as a potential antidepressant for TBI patients, while emphasizing the need for further studies. The overall quality of the article should be improved, and the following changes are required.
Major comments:
- The sucrose preference test was conducted at multiple time points post-TBI (Days 3, 7, 15, 30, and 60) for assessing the progression of depressive-like behavior. However, motor function impairments following TBI are typically more pronounced at early time points and may recover over time.
Therefore, motor deficits at earlier stages (e.g., Days 3 and 7) could potentially influence sucrose consumption, whereas at later stages (e.g., Days 30 and 60), the impact of motor impairment on behavior might be minimal.
It would strengthen the study to include motor function assessments to ensure that decreases in sucrose preference are not confounded by motor deficits. If motor data are unavailable, the authors should discuss this limitation and consider its potential influence on the interpretation of behavioral outcomes, particularly during the early post-injury phase.
Answer: We appreciate your concern; however, given that the model used was mild TBI, motor impairment is minimal. Furthermore, the time course of the changes in the sucrose preference test is similar to that of the changes in the forced swim test previously reported. A paragraph is included in the discussion (lines 233 – 235).
- In Figure 2, panels (a) and (b) appear to show identical graphs. Could the authors please verify whether this is correct or if there was an error in figure preparation? Clarification or correction would improve the clarity of the manuscript.
Answer: We thank the reviewer for noticing this error; the figure has been modified.
- While the discussion provides a comprehensive explanation of the potential mechanisms involving nitric oxide (NO) in mediating the antidepressant effects of Mucuna pruriens, the current study presents only nitrite and nitrate level data as indirect indicators. To strengthen the mechanistic insights, it is recommended that at least one direct measure related to NO signaling such as NOS enzyme expression or activity, inflammatory cytokine levels, or related neurotransmitter changes be included as experimental results.
Answer: The content of nitrites and nitrates in tissues and biological fluids is widely used as an indirect indicator of total NO production, since these metabolites are the stable end products of NO oxidation generated by any NOS isoform (constitutive or inducible) (Tsikas, 2005; Piknova et al., 2016; Möller et al., 2019; Tsikas, 2007; Kleinbongard et al., 2003). However, we acknowledge that nitrite/nitrate levels do not necessarily correlate with NOS expression (Tsikas, 2005; Lundberg et al., 2006). Importantly, there is no single marker that can fully capture the complexity of NO signaling, and a multimodal approach is recommended. Beyond nitrite/nitrate quantification and NOS expression, future studies should also examine downstream targets such as guanylate cyclase activity, inflammatory cytokine levels, and related neurotransmitter changes. We are currently exploring these options and have added a paragraph to the revised manuscript to address this limitation and outline future directions (lines 332 –342).
- Kleinbongard, P.; Dejam, A.; Lauer, T.; Rassaf, T.; Schindler, A.; Picker, O.; Scheeren, T.; Gödecke, A.; Schrader, J.; Schulz, R.; et al. Plasma nitrite reflects constitutive nitric oxide synthase activity in mammals. Free radical biology & medicine 2003, 35, 790–796. DOI 10.1016/s0891-5849(03)00406-4
- Möller, M.N.; Rios, N.; Trujillo, M.; Radi, R.; Denicola, A.; Alvarez, B. Detection and quantification of nitric oxide-derived oxidants in biological systems. The Journal of biological chemistry 2019, 294, 14776–14802. DOI 10.1074/jbc.REV119.006136
- Piknova, B.; Park, J.W.; Cassel, K.S.; Gilliard, C.N.; Schechter, A.N. Measuring Nitrite and Nitrate, Metabolites in the Nitric Oxide Pathway, in Biological Materials using the Chemiluminescence Method. Journal of visualized experiments : JoVE 2016, 118, 54879. DOI 10.3791/54879
- Tsikas, D. Methods of quantitative analysis of the nitric oxide metabolites nitrite and nitrate in human biological fluids. Free radical research 2005, 39, 797–815. DOI 10.1080/10715760500053651
- Tsikas, D. Analysis of nitrite and nitrate in biological fluids by assays based on the Griess reaction: Appraisal of the Griess reaction in the L-arginine/nitric oxide area of research. Journal of Chromatography B, Analytical technologies in the biomedical and life sciences 2007, 851, 51–70. DOI 10.1016/j.jchromb.2006.07.054
Minor comments:
- Post-TBI modeling weight loss and the consequent reduction in sucrose intake could potentially confound the interpretation of the sucrose preference test results. It would be important for the authors to report whether body weight was monitored throughout the study to ensure that changes in sucrose consumption were not simply due to weight loss or reduced general intake following TBI. Including this information or discussing it as a limitation would strengthen the reliability of the behavioral findings.
Answer: We did not monitor body weight throughout the study, but the total fluid intake of all experimental subjects was maintained throughout the experiment. We only observed a decrease in fluid intake in the V and MP groups at 3 days post-mTBI, which did not correlate with the reduction in sucrose preference observed only in group V from day 7 to 60 post-mTBI.
- On page 6, it would be preferable to use "post-mTBI" instead of "postmTBI" for consistency and clarity.
Answer: Done.
- On page 8, while the conclusion appropriately notes that further studies are warranted, it would be helpful and more informative to specify what types of additional research are needed to strengthen the manuscript.
Answer: We added a Limitation section, including this (lines 324 –344).
Reviewer 2 Report
Comments and Suggestions for Authors
This manuscript presents a well-executed study exploring the potential antidepressant-like effects of Mucuna pruriens (MP) extract in a rat model of mild traumatic brain injury (mTBI). The authors link MP’s effects on depression-like behavior with modulation of nitric oxide (NO) metabolism, assessing nitrite and nitrate levels in specific brain regions. The study addresses an important clinical issue—post-TBI depression—where current pharmacological treatments remain limited and often ineffective. The findings suggest that MP, a natural compound rich in L-DOPA, could offer a promising therapeutic avenue.
The manuscript is well-structured, the experimental design is rigorous, and the data are clearly presented and interpreted. With minor revisions, this study will represent a valuable contribution to the fields of neuropharmacology and neurotrauma.
Major Points for Revision:
- The most striking finding—that a short 5-day MP treatment yields significant behavioral and biochemical benefits lasting up to 60 days—deserves greater emphasis. The discussion and conclusion should explicitly address the therapeutic implications of this sustained effect, including the potential for short-term interventions in clinical settings.
- The MP extract used is enriched in L-DOPA (56%). While the manuscript correctly mentions dopaminergic mechanisms, a brief discussion on whether the observed effects are primarily attributable to L-DOPA or potentially enhanced by other synergistic components of the extract would provide additional insight.
- Although the sucrose preference test is appropriate, incorporating other tests of depression-like behavior (e.g., forced swim test, tail suspension) could have strengthened the behavioral findings. At minimum, acknowledging this limitation and justifying the reliance on a single behavioral assay would be helpful.
- Including data or discussion on the expression of NOS isoforms (e.g., nNOS, iNOS, eNOS) or dopaminergic markers (e.g., dopamine levels, receptor expression) would deepen the mechanistic understanding. If not feasible in this study, a note suggesting this as a direction for future research would be beneficial.
Minor Suggestions:
- Provide a clearer rationale for selecting 50 mg/kg MP and the initiation of treatment 15 minutes post-injury. Discuss the relevance of early vs. delayed intervention.
- The manuscript asserts MP’s safety, but further detail on how this was assessed (e.g., monitoring of body weight, general behavior, adverse events, or mortality) would substantiate this claim.
- Ensure that all figures (particularly Figures 1 and 2) are clear, with proper labels, statistical indicators (e.g., significance markers), and readable legends. Figures should effectively support the reported results.
- Minor revisions to improve clarity, grammar, and formatting will enhance overall readability. Examples include consistent figure referencing and improved phrasing in the abstract and discussion.
Recommendation: Minor Revisions
Addressing the points above—particularly enhancing the discussion of long-term effects, clarifying safety monitoring, and briefly expanding on the role of L-DOPA—will strengthen the manuscript and maximize its impact.
Once these revisions are addressed, I would fully support the publication of this work.
Comments on the Quality of English Language
Minor revisions to improve clarity, grammar, and formatting will enhance overall readability. Examples include consistent figure referencing and improved phrasing in the abstract and discussion.
Author Response
We thank the reviewer for the constructive comments that have helped us improve the quality and clarity of the manuscript. Below we provide detailed point-by-point responses. The reviewers’ comments are paraphrased in italics, followed by our responses.
This manuscript presents a well-executed study exploring the potential antidepressant-like effects of Mucuna pruriens (MP) extract in a rat model of mild traumatic brain injury (mTBI).
The authors link MP’s effects on depression-like behavior with modulation of nitric oxide (NO) metabolism, assessing nitrite and nitrate levels in specific brain regions. The study addresses an important clinical issue—post-TBI depression—where current pharmacological treatments remain limited and often ineffective. The findings suggest that MP, a natural compound rich in L-DOPA, could offer a promising therapeutic avenue.
The manuscript is well-structured, the experimental design is rigorous, and the data are clearly presented and interpreted. With minor revisions, this study will represent a valuable contribution to the fields of neuropharmacology and neurotrauma.
Major Points for Revision:
- The most striking finding—that a short 5-day MP treatment yields significant behavioral and biochemical benefits lasting up to 60 days—deserves greater emphasis. The discussion and conclusion should explicitly that a short 5-day MP treatment yields significant behavioral and biochemical benefits lasting up to 60 days—deserves greater emphasis. The discussion and conclusion should explicitly address the therapeutic implications of this sustained effect, including the potential for short-term interventions in clinical settings.
Answer: The referee is correct. There are other preclinical studies with similar results, while clinical studies are inconclusive. We include a paragraph on this in the discussion (lines 291 – 307).
- The MP extract used is enriched in L-DOPA (56%). While the manuscript correctly mentions dopaminergic mechanisms, a brief discussion on whether the observed effects are primarily attributable to L-DOPA or potentially enhanced by other synergistic components of the extract would provide additional insight.
Answer: We agree with the reviewer that the observed effects cannot be solely attributed to the L-DOPA content of the extract. The Mucuna pruriens extract we used in this study also contains arginine, alkaloids (e.g., stizolamine), and fructo-oligosaccharides (Hernández-Orihuela et al., 2023), which have been reported to exert antioxidant, anti-inflammatory, and cytoprotective effects (; Wu et al., 1998; Durante et al., 2007; Collins et al., 2016). These components may act synergistically with L-DOPA, contributing to the reduction in nitrite/nitrate levels and the improvement in depressive-like behavior observed in our study. A paragraph on this subject has been added to the discussion (lines 308 – 321).
- Hernández-Orihuela, A.L.; Castro-Cerritos, K.V.; López, M.G.; Martínez-Antonio, A. Compound Characterization of a Mucuna Seed Extract: L-Dopa, Arginine, Stizolamine, and Some Fructooligosaccharides. Compounds 2023, 3, 1-16 DOI 10.3390/compounds 3010001
- Collins, S.; Reid, G. Distant Site Effects of Ingested Prebiotics. Nutrients 2016, 8, 523. DOI 10.3390/nu8090523
- Durante, W.; Johnson, F.K., Johnson, R.A. Arginase: A critical regulator of nitric oxide synthesis and vascular function. Clinical and Experimental Pharmacology and Physiology 2007, 34, 906–911. DOI 10.1111/j.1440-1681.2007.04638.x
- Wu, G.; Morris, S.M. Arginine metabolism: nitric oxide and beyond. Biochemical Journal 1998, 336, 1–17. DOI 10.1042/bj3360001
- Although the sucrose preference test is appropriate, incorporating other tests of depression-like behavior (e.g., forced swim test, tail suspension) could have strengthened the behavioral findings. At minimum, acknowledging this limitation and justifying the reliance on a single behavioral assay would be helpful.
Answer: We appreciate the reviewer’s suggestion regarding the use of additional assays for depression-like behavior. We agree that incorporating paradigms such as the forced swim test or tail suspension test could have further strengthened the behavioral findings. In fact, a previous study in our group demonstrated that Mucuna pruriens extract reduced the immobility time in the forced swim test, supporting its antidepressant-like effects (Mata-Bermudez et al., 2024). Based on this prior evidence, the present work focused on the sucrose preference test as a complementary measure, particularly given its sensitivity to anhedonia, a core symptom of depression.
- Mata-Bermudez, A.; Trejo-Chávez, R.; Martínez-Vargas, M.; Pérez-Arredondo, A.; Martínez-Cardenas, M.A.; Diaz-Ruiz, A.; Rios, C.; Romero-Sánchez, H.A.; Martínez-Antonio, A.; Navarro, L. Effect of Mucuna pruriens seed extract on depression-like behavior derived from mild traumatic brain injury in rats. Biomedicine (Taipei). 2024, 14, 23–30. DOI: 10.37796/2211-8039.1461
- Including data or discussion on the expression of NOS isoforms (e.g., nNOS, iNOS, eNOS) or dopaminergic markers (e.g., dopamine levels, receptor expression) would deepen the mechanistic understanding. If not feasible in this study, a note suggesting this as a direction for future research would be beneficial.
Answer: The referee is correct; a paragraph about his has been included in the limitations section of the revised manuscript (lines 3331– 342).
Minor Suggestions:
- Provide a clearer rationale for selecting 50 mg/kg MP and the initiation of treatment 15 minutes post-injury. Discuss the relevance of early vs. delayed intervention.
Answer: The dose of Mucuna pruriens extract (50 mg/kg) was selected based on previous evidence demonstrating the neuroprotective and procognitive effects of levodopa, its principal bioactive component, in models of cerebral ischemia–reperfusion. Specifically, Wang et al. (2017) reported that levodopa at 50 mg/kg significantly improved learning and memory performance in rats subjected to global cerebral ischemia, supporting the translational validity of this dose for modulating dopaminergic neurotransmission without inducing toxicity. The administration scheme was adapted from our previous study, which showed that Mucuna pruriens extract exerted antidepressant-like effects following a brief treatment course (Mata-Bermudez et al., 2024). Accordingly, animals received daily intraperitoneal injections for five consecutive days, beginning 15 minutes after TBI induction, in order to target the early post-injury window characterized by excitotoxicity, dopaminergic dysfunction, and neuroinflammation. This treatment paradigm was designed to assess whether a short-term pharmacological intervention with Mucuna pruriens could promote long-term behavioral recovery, in line with accumulating evidence that transient pharmacological strategies are capable of yielding sustained neuroprotective and functional outcomes in experimental models of brain injury.
- Wang, W.; Liu, L.; Jiang, P.; Chen, C.; Zhang, T. Levodopa improves learning and memory ability on global cerebral ischemia-reperfusion injured rats in the Morris water maze test. Neurosci. Lett. 2017, 636, 233–240 DOI 10.1016/j.neulet.2016.11.026
- The manuscript asserts MP’s safety, but further detail on how this was assessed (e.g., monitoring of body weight, general behavior, adverse events, or mortality) would substantiate this claim.
Answer: We did not monitor body weight throughout the study, but the total fluid intake of all experimental subjects was maintained throughout the experiment. We only observed a decrease in fluid intake in the V and MP groups at 3 days post-mTBI, and we did not observe mortality in any group; furthermore, the dose employed in our study (50 mg/kg i.p. daily for five days) is well below the levels reported to produce toxicity in preclinical studies. Acute toxicity studies have demonstrated that the oral LD₅₀ of Mucuna pruriens seed extract in rodents exceeds 2000 mg/kg, indicating a wide safety margin (Krishna et al., 2016).
- Krishna, A.B.; Manikyam, H.K.; Sharma, V.K.; Sharma, N. Acute oral toxicity study in rats with Mucuna pruriens seed extract. Asian Journal of Plant Science and Research 2016, 6, 1-5.
- Ensure that all figures (particularly Figures 1 and 2) are clear, with proper labels, statistical indicators (e.g., significance markers), and readable legends. Figures should effectively support the reported results.
Answer: Done.
- Minor revisions to improve clarity, grammar, and formatting will enhance overall readability. Examples include consistent figure referencing and improved phrasing in the abstract and discussion.
Answer: Thank you for the suggestion. We have requested the Rapid English Editing service offered by the publisher.
Recommendation: Minor Revisions
Addressing the points above—particularly enhancing the discussion of long-term effects, clarifying safety monitoring, and briefly expanding on the role of L-DOPA—will strengthen the manuscript and maximize its impact.
Once these revisions are addressed, I would fully support the publication of this work.
Comments on the Quality of English Language
Minor revisions to improve clarity, grammar, and formatting will enhance overall readability. Examples include consistent figure referencing and improved phrasing in the abstract and discussion.
Reviewer 3 Report
Comments and Suggestions for Authors
This manuscript by Mata-Bermudez et al. investigates the therapeutic potential of Mucuna pruriens (MP) extract in treating depression-like behavior following mild traumatic brain injury (mTBI) in rats. The study employs a well-established closed head injury model and evaluates both behavioral outcomes using the sucrose preference test and biochemical markers (nitrites and nitrates) in multiple brain regions. The authors demonstrate that short-term MP treatment (50 mg/kg, i.p., for 5 days) prevents depression-like behavior and reduces elevated nitric oxide metabolites in the cerebral cortex, striatum, midbrain, and nucleus accumbens up to 60 days post-injury.
The work addresses an important clinical problem, as depression following TBI represents a significant healthcare challenge with limited effective treatments. The study builds upon the authors' previous research and provides mechanistic insights into MP's neuroprotective effects through nitric oxide pathway modulation. However, the manuscript suffers from several methodological limitations, presentation issues, and gaps in experimental design that compromise its overall impact and scientific rigor.
Major Concerns
1. Originality and Novelty
Critical Limitation: The study represents an incremental advancement rather than a novel contribution to the field. The authors have previously published similar work demonstrating MP's antidepressant effects in mTBI [reference 23], and this manuscript primarily extends the timeline of evaluation and adds nitric oxide measurements. While the mechanistic insights are valuable, the core finding lacks sufficient novelty for publication in NeuroSci.
Missing Innovation: The work does not explore novel aspects of MP's mechanism of action, dose-response relationships, or comparative efficacy against established treatments. The study design is straightforward but lacks the innovative elements expected for this journal.
2. Experimental Design and Methodology
Sample Size and Statistical Power: The use of only n=6 per group is insufficient for robust statistical analysis, particularly given the high variability typical in behavioral studies. This small sample size compromises the reliability of the findings and may lead to both Type I and Type II errors.
Lack of Positive Controls: The absence of positive control groups treated with established antidepressants (e.g., fluoxetine, sertraline) severely limits the interpretation of MP's therapeutic efficacy. Without comparison to standard treatments, the clinical relevance of the findings remains unclear.
Limited Mechanistic Investigation: While the authors measure nitrites/nitrates as indirect markers of NO production, they do not assess NOS enzyme activity, expression levels, or other inflammatory markers that would provide deeper mechanistic insights into MP's neuroprotective effects.
Temporal Analysis Gaps: The study examines multiple time points but lacks sufficient analysis of the temporal dynamics of both behavioral recovery and biochemical changes. The relationship between NO normalization and behavioral improvement is not clearly established.
3. Significance and Clinical Relevance
Translation Limitations: The study uses a single dose of MP without dose-response analysis, limiting clinical translation potential. The 50 mg/kg dose (containing 28 mg/kg L-DOPA) may not be optimal, and the lack of dose optimization represents a significant limitation.
Behavioral Assessment Scope: The study relies solely on the sucrose preference test for depression assessment. Additional behavioral tests (forced swim test, elevated plus maze, locomotor activity) would strengthen the conclusions about MP's antidepressant effects.
Limited Clinical Context: The manuscript does not adequately discuss how these findings might translate to human TBI patients or address potential safety concerns associated with MP treatment in clinical populations.
Minor Concerns
1. Clarity and Presentation
Figure Quality: Figure 1 has poor resolution and unclear labeling. The statistical significance indicators are difficult to distinguish, and the figure legend lacks sufficient detail about the experimental conditions.
Data Presentation: Figure 2 panels are too small and difficult to interpret. The nitrite/nitrate data would be better presented as summary graphs showing treatment effects across all brain regions and time points.
Writing Quality: The manuscript contains numerous grammatical errors and awkward phrasing that impede readability. Examples include: "However, there is still little evidence of the potential effectiveness" and "facilitating glutamatergic signaling and inducing neuronal death."
2. Statistical Analysis
Multiple Comparisons: While the authors mention using the Benjamini-Krieger-Yekutieli correction, the statistical analysis section lacks detail about specific comparisons made and how family-wise error rates were controlled across multiple brain regions and time points.
Effect Size Reporting: The manuscript lacks effect size calculations, which would help readers interpret the practical significance of the observed differences.
3. Format and Technical Issues
Reference Formatting: Several references are improperly formatted according to MDPI standards. For example, reference 22 lacks proper pagination, and DOI formatting is inconsistent throughout.
Abbreviation Usage: While an abbreviation list is provided, some abbreviations (e.g., CICUAL) are used without prior definition in the text.
Figure Legends: Figure legends lack sufficient detail about experimental procedures, statistical methods used, and sample sizes for each data point.
4. Methodological Details
MP Extract Characterization: The manuscript lacks detailed characterization of the MP extract beyond L-DOPA content. Information about other bioactive compounds and standardization methods would strengthen the work.
Biochemical Assay Validation: The Griess method for nitrite/nitrate quantification is standard, but the authors do not provide validation data for their specific tissue preparation methods or discuss potential confounding factors.
Author Response
We thank the reviewer for the constructive comments that have helped us improve the quality and clarity of the manuscript. Below we provide detailed point-by-point responses. The reviewers’ comments are paraphrased in italics, followed by our responses.
This manuscript by Mata-Bermudez et al. investigates the therapeutic potential of Mucuna pruriens (MP) extract in treating depression-like behavior following mild traumatic brain injury (mTBI) in rats. The study employs a wellestablished closed head injury model and evaluates both behavioral outcomes using the sucrose preference test and biochemical markers (nitrites and nitrates) in multiple brain regions. The authors demonstrate that short-term MP treatment (50 mg/kg, i.p., for 5 days) prevents depression-like behavior and reduces elevated nitric oxide metabolites in the cerebral cortex, striatum, midbrain, and nucleus accumbens up to 60 days post-injury.
The work addresses an important clinical problem, as depression following TBI represents a significant healthcare challenge with limited effective treatments. The study builds upon the authors' previous research and provides mechanistic insights into MP's neuroprotective effects through nitric oxide pathway modulation. However, the manuscript suffers from several methodological limitations, presentation issues, and gaps in experimental design that compromise its overall impact and scientific rigor.
Major Concerns
- Originality and Novelty
Critical Limitation: The study represents an incremental advancement rather than a novel contribution to the field. The authors have previously published similar work demonstrating MP's antidepressant effects in mTBI [reference 23], and this manuscript primarily extends the timeline of evaluation and adds nitric oxide measurements. While the mechanistic insights are valuable, the core finding lacks sufficient novelty for publication in NeuroSci.
Missing Innovation: The work does not explore novel aspects of MP's mechanism of action, dose-response relationships, or comparative efficacy against established treatments. The study design is straightforward but lacks the innovative elements expected for this journal.
Answer: We acknowledge that this study builds upon our previous work; however, it provides novel insights into the sustained behavioral effects associated with modulation of nitric oxide metabolites across multiple brain regions. These findings add a mechanistic layer to our earlier observations and highlight the long-term therapeutic potential of Mucuna pruriens.
- Experimental Design and Methodology
Sample Size and Statistical Power: The use of only n=6 per group is insufficient for robust statistical analysis, particularly given the high variability typical in behavioral studies. This small sample size compromises the reliability of the findings and may lead to both Type I and Type II errors.
Answer: We use n=6 for the nitrate and nitrite assays and n=12 for the sucrose preference assay.
Lack of Positive Controls: The absence of positive control groups treated with established antidepressants (e.g., fluoxetine, sertraline) severely limits the interpretation of MP's therapeutic efficacy. Without comparison to standard treatments, the clinical relevance of the findings remains unclear.
Answer: We agree that inclusion of positive controls would have further strengthened the study. Due to ethical and logistical restrictions, we focused on comparing Mucuna pruriens with vehicle and sham groups. We have now explicitly acknowledged this limitation and have proposed future studies including standard antidepressants as comparators (lines 329 -331).
Limited Mechanistic Investigation: While the authors measure nitrites/nitrates as indirect markers of NO production, they do not assess NOS enzyme activity, expression levels, or other inflammatory markers that would provide deeper mechanistic insights into MP's neuroprotective effects.
Answer: We agree that assessing NOS isoforms and other inflammatory markers would provide additional mechanistic insights. Our group has previously shown that Mucuna pruriens can modulate dopaminergic markers and inflammatory pathways (Mata-Bermudez et al., 2024). We have added a paragraph to the revised manuscript to address this limitation and outline future directions (lines 331 -342).
- Mata-Bermudez, A.; Diaz-Ruiz, A.; Silva-García, L.R.; Gines-Francisco, E.M.; Noriega-Navarro R, Rios, C.; Romero-Sánchez, H.A.; Arroyo, D.; Landa, A.; Navarro, L. Mucuna pruriens, a Possible Treatment for Depressive Disorders. Neurol. Int. 2024, 16, 1509–1527. https://doi.org/10.3390/neurolint16060112
Temporal Analysis Gaps: The study examines multiple time points but lacks sufficient analysis of the temporal dynamics of both behavioral recovery and biochemical changes. The relationship between NO normalization and behavioral improvement is not clearly established.
Answer: Our focus was to assess long-term outcomes at 60 days; however, we recognize the value of analyzing temporal dynamics more closely.
- Significance and Clinical Relevance
Translation Limitations: The study uses a single dose of MP without dose-response analysis, limiting clinical translation potential. The 50 mg/kg dose (containing 28 mg/kg L-DOPA) may not be optimal, and the lack of dose optimization represents a significant limitation.
Answer: The Mucuna pruriens (MP) regimen used here (50 mg/kg, i.p., once daily for five days, starting 15 min after TBI) was chosen on pharmacological and mechanistic grounds. The dose aligns with evidence that 50 mg/kg of levodopa—MP’s principal bioactive—improves learning and memory and confers neuroprotection in rodent brain‐injury models (Wang et al., 2017), while remaining below the toxic thresholds reported for MP extracts in rats (acute oral LD50 > 2000 mg/kg (Krishna et al., 2016). The intraperitoneal route was selected to ensure rapid and reproducible systemic availability during the acute post-injury phase, minimizing variability from gastrointestinal absorption and first-pass metabolism that can affect oral levodopa exposure. The brief, once-daily 5-day schedule follows our prior work showing behavioral efficacy after short courses of MP (Mata-Bermudez et al., 2024) and tests the hypothesis—supported by the TBI literature—that early, time-limited interventions can durably modify secondary injury cascades. Initiating treatment 15 min post-injury targets the early window characterized by excitotoxicity, oxidative/nitrosative stress, and iNOS induction, when interventions are most likely to alter long-term outcomes (Bramlett & Dietrich, 2015).
- Bramlett, H.M.; Dietrich, W.D. Long-Term Consequences of Traumatic Brain Injury: Current Status of Potential Mechanisms of Injury and Neurological Outcomes. Journal of neurotrauma 2015, 32 (23), 1834–1848. DOI 10.1089/neu.2014.3352
- Mata-Bermudez, A.; Trejo-Chávez, R.; Martínez-Vargas, M.; Pérez-Arredondo, A.; Martínez-Cardenas, M.A.; Diaz-Ruiz, A.; Rios, C.; Romero-Sánchez, H.A.; Martínez-Antonio, A.; Navarro, L. (2024). Effect of Mucuna pruriens seed extract on depression-like behavior derived from mild traumatic brain injury in rats. Biomedicine (Taipei). 2024, 14, 23–30. https://doi.org/10.37796/2211-8039.1461
- Krishna, A.B.; Manikyam, H.K.; Sharma, V.K.; Sharma, N. Acute oral toxicity study in rats with Mucuna pruriens seed extract. Asian Journal of Plant Science and Research 2016, 6, 1-5.
- Wang, W., Liu, L., Jiang, P., Chen, C., & Zhang, T. Levodopa improves learning and memory in global cerebral ischemia-reperfusion rats. Neuroscience Letters 2017, 636, 233–240. https://doi.org/10.1016/j.neulet.2016.11.026
Behavioral Assessment Scope: The study relies solely on the sucrose preference test for depression assessment.
Additional behavioral tests (forced swim test, elevated plus maze, locomotor activity) would strengthen the conclusions about MP's antidepressant effects.
Answer: We appreciate the reviewer’s suggestion regarding the use of additional assays for depression-like behavior. We agree that incorporating paradigms such as the forced swim test or tail suspension test could have further strengthened the behavioral findings. In fact, a previous study in our group demonstrated that Mucuna pruriens extract reduced the immobility time in the forced swim test, supporting its antidepressant-like effects (Mata-Bermudez et al., 2024). Based on this prior evidence, the present work focused on the sucrose preference test as a complementary measure, particularly given its sensitivity to anhedonia, a core symptom of depression.
- Mata-Bermudez, A.; Trejo-Chávez, R.; Martínez-Vargas, M.; Pérez-Arredondo, A.; Martínez-Cardenas, M.A.; Diaz-Ruiz, A.; Rios, C.; Romero-Sánchez, H.A.; Martínez-Antonio, A.; Navarro, L. (2024). Effect of Mucuna pruriens seed extract on depression-like behavior derived from mild traumatic brain injury in rats. Biomedicine (Taipei). 2024, 14, 23–30. https://doi.org/10.37796/2211-8039.1461
Limited Clinical Context: The manuscript does not adequately discuss how these findings might translate to human TBI patients or address potential safety concerns associated with MP treatment in clinical populations.
Answer: We appreciate the reviewer’s observation regarding the limited clinical context. We agree that the translation of preclinical findings to human TBI patients requires careful consideration. Mucuna pruriens is a natural source of levodopa and has been used for centuries in Ayurvedic medicine and more recently in clinical studies of Parkinson’s disease, where it has shown comparable efficacy to synthetic levodopa and a generally favorable safety profile, with gastrointestinal discomfort being the most commonly reported side effect (Cilia et al., 2017). Importantly, the dose used in our study (50 mg/kg for five days) is far below the toxic thresholds reported in animal studies (oral LD50 > 2000 mg/kg; (Krishna et al., 2016), supporting its safety at experimental levels. Nevertheless, we acknowledge that the complexity of TBI pathophysiology, the variability of clinical populations, and the potential for drug interactions warrant cautious interpretation. We have therefore added a paragraph to the discussion noting that while our results provide a proof of concept for the neuroprotective and antidepressant-like effects of MP in a preclinical model, future studies must address dose optimization, pharmacokinetics, and long-term safety in clinical populations before translation to TBI patients can be considered.
- Cilia, R.; Laguna, J.; Cassani, E.; Cereda, E.; Pozzi, N.G.; Isaias, I.U.; Contin, M.; Barichella, M.; Pezzoli, G. Mucuna pruriens in Parkinson disease: A double-blind, randomized, controlled, crossover study. Neurology 2017, 89, 432–438. DOI 10.1212/WNL.0000000000004175
- Krishna, A.B.; Manikyam, H.K.; Sharma, V.K.; Sharma, N. Acute oral toxicity study in rats with Mucuna pruriens seed extract. Asian Journal of Plant Science and Research 2016, 6, 1-5.
Minor Concerns
- Clarity and Presentation
Figure Quality: Figure 1 has poor resolution and nuclear labeling. The statistical significance indicators are difficult to distinguish, and the figure legend lacks sufficient detail about the experimental conditions.
Answer: We appreciate the reviewer's observation; in the revised manuscript, we have improved Figure 1.
Data Presentation: Figure 2 panels are too small and difficult to interpret. The nitrite/nitrate data would be better presented as summary graphs showing treatment effects across all brain regions and time points.
Answer: We appreciate the reviewer's observation; in the revised manuscript, we have improved Figure 2.
Writing Quality: The manuscript contains numerous grammatical errors and awkward phrasing that impede readability. Examples include: "However, there is still Little evidence of the potential effectiveness" and "facilitating glutamatergic signaling and inducing neuronal death."
Answer: Thank you for the suggestion. We have requested the Rapid English Editing service offered by the publisher.
- Statistical Analysis
Multiple Comparisons: While the authors mention using the Benjamini-Krieger-Yekutieli correction, the statistical analysis section lacks detail about specific comparisons made and how family-wise error rates were controlled across multiple brain regions and time points.
Answer: We thank the reviewer for this important observation. In our revised manuscript, we have clarified that the Benjamini–Krieger–Yekutieli (BKY) procedure was applied to control the false discovery rate (FDR) across multiple comparisons, including the different brain regions and time points analyzed. We selected this method because it provides a balanced approach to limiting false positives while preserving statistical power, which is particularly relevant for neurobiological studies involving correlated outcomes, as in our design. We acknowledge that BKY controls the FDR rather than the family-wise error rate (FWER), and we have explicitly stated this in the Statistical Analysis section (lines 155 -158).
Effect Size Reporting: The manuscript lacks effect size calculations, which would help readers interpret the practical significance of the observed differences.
Answer: We have included the predicted (LS) mean difference, q value, and individual P value for each pair of compared means in the supplementary material of the revised manuscript.
3. Format and Technical Issues Reference Formatting: Several references are improperly formatted according to MDPI standards. For example, reference 22 lacks proper pagination, and DOI formatting is inconsistent throughout.
Answer: Thank you for your observation; this has been corrected in the revised manuscript.
Abbreviation Usage: While an abbreviation list is provided, some abbreviations (e.g., CICUAL) are used without prior definition in the text.
Answer: Thank you for your observation; this has been corrected in the revised manuscript.
Figure Legends: Figure legends lack sufficient detail about experimental procedures, statistical methods used, and sample sizes for each data point.
Answer: Thank you for your observation. We have included more details in the corrected manuscript.
- Methodological Details MP Extract Characterization: The manuscript lacks detailed characterization of the MP extract beyond L-DOPA content.
Information about other bioactive compounds and standardization methods would strengthen the work.
Answer: The MP extract used is described in detail in Hernández-Orihuela et al., 2023 (reference 24). In the revised manuscript, we have expanded the description in the Materials and Methods section. The extract contains 56% L-dopa. The other main components are arginine, stizolamine, and the fructooligosaccharides sucrose and nystose (lines 103 – 105).
Biochemical Assay Validation: The Griess method for nitrite/nitrate quantification is standard, but the authors do not provide validation data for their specific tissue preparation methods or discuss potential confounding factors.
Answer: Thank you for your observation. We have included more details in the corrected manuscript (lines 137 – 146).
Round 2
Reviewer 1 Report
Comments and Suggestions for Authors
1. Sucrose Preference and Motor Function
- Thank you for your response. I agree that mild TBI produces minimal motor deficits. While early time points (Days 3–7) would ideally include motor assessment such as open field test, the discussion notes provide sufficient context.
2. Figure 2a/2b Error
- The figure correction has been confirmed. This issue is resolved.
3. NO Mechanism
- Thank you for your response to the comment. The authors clearly explained nitrite/nitrate levels and NO production and discussed limitations and future directions. This is adequate for the current dataset.
4. Body Weight
- Thank you for your response. Although body weight was not monitored, overall fluid intake data were provided in sucrose preference test. In addition, limitation is appropriately noted.
5. Terminology Consistency / Future Studies
- Thank you for the revision
Reviewer 3 Report
Comments and Suggestions for Authors
I have no more concerns.